# Diagnostic and Prognostic Power of Active DNA Demethylation Pathway Intermediates in Acute Myelogenous Leukemia and Myelodysplastic Syndromes

**DOI:** 10.3390/cells11050888

**Published:** 2022-03-04

**Authors:** Aleksandra Skalska-Bugala, Marta Starczak, Łukasz Szukalski, Maciej Gawronski, Agnieszka Siomek-Gorecka, Justyna Szpotan, Anna Labejszo, Ewelina Zarakowska, Anna Szpila, Anna Jachalska, Adriana Szukalska, Marcin Kruszewski, Anna Sadowska, Aleksandra Wasilow, Patrycja Baginska, Jaroslaw Czyz, Ryszard Olinski, Rafal Rozalski, Daniel Gackowski

**Affiliations:** 1Department of Clinical Biochemistry, Faculty of Pharmacy, Collegium Medicum in Bydgoszcz, Nicolaus Copernicus University in Toruń, 85-092 Bydgoszcz, Poland; aleksandraaskalska@gmail.com (A.S.-B.); marta.starczak@cm.umk.pl (M.S.); m.gawronski@cm.umk.pl (M.G.); asiomek@cm.umk.pl (A.S.-G.); justyna.szpotan@umk.pl (J.S.); anna.labejszo@cm.umk.pl (A.L.); ewelinaz@cm.umk.pl (E.Z.); szpila@cm.umk.pl (A.S.); aleksandra.wasilow@gmail.com (A.W.); patrycja.baginska.poczta@gmail.com (P.B.); ryszardo@cm.umk.pl (R.O.); 2Department of Hematology, Faculty of Medicine, Collegium Medicum in Bydgoszcz, Nicolaus Copernicus University in Toruń, 85-168 Bydgoszcz, Poland; lukasz.szukalski@cm.umk.pl (Ł.S.); anna.jachalska@cm.umk.pl (A.J.); jczyz@cm.umk.pl (J.C.); 3Department of Human Biology, Institute of Biology, Faculty of Biological and Veterinary Sciences, Nicolaus Copernicus University in Toruń, 87-100 Toruń, Poland; 4Department of Geriatrics, Division of Biochemistry and Biogerontology, Collegium Medicum in Bydgoszcz, Nicolaus Copernicus University in Toruń, 85-094 Bydgoszcz, Poland; 5Clinic of Hematology, University Hospital No. 2—Jan Biziel Memorial Hospital, 85-168 Bydgoszcz, Poland; adriana.czyz@hotmail.co.uk (A.S.); marcin.kruszewski5@wp.pl (M.K.); 6Department of Hematology, Nicolaus Copernicus Hospital, 87-100 Toruń, Poland; an.sadowska94@gmail.com

**Keywords:** acute myelogenous leukemia, myelodysplastic syndromes, biomarkers, active demethylation, 5-hydroxymethylcytosine, 5-carboxycytosine, 5-formylcytosine, 5-hydroxymethyluracil, ROC, classification trees

## Abstract

Acute myeloid leukemia (AML) and myelodysplastic syndromes (MDS) are characterized by genomic instability, which may arise from the global hypomethylation of the DNA. The active DNA demethylation process may be linked with aberrant methylation and can be involved in leukemogenesis. The levels of 5-methylcytosine oxidation products were analyzed in minimally invasive material: the cellular DNA from peripheral blood cells and urine of patients with AML and MDS along with the control group, using isotope-dilution two-dimensional ultra-performance liquid chromatography with tandem mass spectrometry. The receiver operating characteristic curve analysis was used for the assessment of the ability to discriminate patients’ groups from the control group, and AML from MDS. The most diagnostically useful for discriminating AML patients from the control group was the urinary excretion of 5-hydroxymethylcytosine (AUC = 0.918, sensitivity: 85%, and specificity: 97%), and 5-(hydroxymethyl)-2′-deoxyuridine (0.873, 74%, and 92%), while for MDS patients 5-(hydroxymethyl)-2′-deoxycytidine in DNA (0.905, 82%, and 98%) and urinary 5-hydroxymethylcytosine (0.746, 66%, and 92%). Multi-factor models of classification trees allowed the correct classification of patients with AML and MDS in 95.7% and 94.7% of cases. The highest prognostic value of the analyzed parameters in predicting the transformation of MDS into AML was observed for 5-carboxy-2′-deoxycytidine (0.823, 80%, and 97%) and 5-(hydroxymethyl)-2′-deoxyuridine (0.872, 100%, and 75%) in DNA. The presented research proves that the intermediates of the active DNA demethylation pathway determined in the completely non-invasive (urine) or minimally invasive (blood) material can be useful in supporting the diagnostic process of patients with MDS and AML. The possibility of an early identification of a group of MDS patients with an increased risk of transformation into AML is of particular importance.

## 1. Introduction

Acute myeloid leukemia (AML) is the most common acute leukemia in adults, in particular, those over 60 years old; it also accounts for 15–20% of cases in children. This genetically heterogeneous disease is characterized by the malignant clonal proliferation of immature myeloid cells in the bone marrow, peripheral blood, and sometimes peripheral organs. Myelodysplastic syndromes (MDS) are a group of various diseases, the most notable feature of which is a reduced number of peripheral blood cells (red blood cells, white blood cells, and/or platelets) due to their abnormal formation in the marrow. Over time, there is a gradual worsening of symptoms, mainly caused by a gradual decline in the number of blood cells, while in some patients, MDS may transform into AML. The prognosis for most AML subtypes is very unfavorable, with an overall 5-year survival rate of about twenty-five percent [1,2].

The diagnosis of MDS is based on the identification of cytopenia of one, two, or three erythroid, granulocytic, or megakaryocytic cell lines and bone marrow cell dysplasia (the detailed morphological classification is based on the 2008 World Health Organization (WHO) version [3] revised in 2016 [4]). The complete evaluation of MDS patients also includes bone marrow cytogenetics [5], immunophenotyping, and sequencing of the genome [6]. Moreover, cellular characteristic in MDS is not stable, and most of the patients acquire additional aberrations, which results in an elevated risk of transformation to AML and a shorter overall survival [7]. According to the WHO guidelines, the current classification of AML relies on cytomorphology, immunophenotyping, cytogenetics, and molecular genetics. Such an interdisciplinary approach is necessary to increase the quality of prognostication and there is still space to introduce new diagnostic parameters [8]. Just recently, Estey et al. recommended defining patients with 10% to 30% blasts as “AML/MDS” to ensure that they would be eligible for either MDS or AML therapies or novel clinical trials [9]. The uncertainty of routine bone marrow aspiration is relatively high, with about 20% of false-negative results reported in MDS and 14% in AML [10]. Approximately 12% discordance in the diagnosis was observed between diagnostic centers [11]. Major limitations in the current diagnostic approach to MDS/AML lay in the limited reproducibility of morphological analysis of dysplasia and in the weak specificity of dysplastic changes [12,13].

The probability of obtaining remission and the risk of cancer recurrence are influenced by the genetic and epigenetic profile of malignant cells. The relationship between many molecular mutations in AML and the prognosis for patients are well known. However, little is known about the effects of mutations in genes with epigenetic functions [14]. The best-known and one of the most important epigenetic modifications is cytosine methylation, which is most commonly observed at CpG dinucleotides. The opposite process, also with epigenetic importance, is the active DNA demethylation. It involves proteins from the Ten Eleven Translocation (TET) family, which oxidize 5-methylcytosine (5-mCyt) to 5-hydroxymethylcytosine (5-hmCyt). 5-HmCyt can be further oxidized to 5-formylcytosine (5-fCyt) and 5-carboxycytosine (5-caCyt) [15,16]. The modified bases described above are then removed from the DNA and replaced with unmodified cytosine via the base excision repair (BER) pathway [17]. Additionally, 5-hydroxymethyluracil (5-hmUra) can be formed by the oxidation of thymine by TET enzymes and can be removed by BER repair. After the excision from the DNA, modified deoxynucleosides and nucleobases appear in the bloodstream and finally may be observed in the urine. There is a common belief that the presence of the modifications in the urine primarily represents the repair product of DNA damage in vivo and reflects the activity of repair pathways [18].

The active DNA demethylation process may activate previously silenced genes, being involved in abnormal methylation, and participating in leukemogenesis. Mutations in genes related to DNA methylation/demethylation pathways (*DNMT3A*, *TET2*, *IDH1*, and *IDH2*) are common in patients with MDS and AML, which affects the DNA modification profile. In 7% to 10% of AML patients, loss-of-function mutations in *TET2* were described; moreover, 10% to 20% of people with AML have heterozygous mutations in *IDH1* and *IDH2* that are mutually exclusive with mutations in *TET2* [19]. More recently, Gurnari et al. found that the depletion of 5-hmCyt and TET2 mRNA are common in MDS irrespective of the *TET2* mutation [20]. In the light of this research “TETopathy”, defined as various factors causing abnormalities in *TETs* expression, activity, or specificity, appears a common feature of MDS/AML patients.

Owing to a plethora of the aforementioned factors suspected of shaping the epigenetic landscape, herein we aimed to investigate the diagnostic value and prognostic power of active DNA demethylation pathway intermediates in AML and MDS. We comprehensively analyzed the levels of 5-mCyt, 5-hmCyt, 5-fCyt, 5-caCyt, and 5-hmUra in the minimally invasive material: the cellular DNA from peripheral blood cells and the urine of patients suffering from AML and its predisposing condition—MDS—along with the control group. Next, we assessed its diagnostic power and identified the most promising biomarkers of AML and MDS development as well as MDS to AML transformation.

## 2. Materials and Methods

Primary, newly diagnosed, treatment-naïve patients with AML and MDS of the Departments of Haematology (Biziel University Hospital No. 2, Collegium Medicum, Bydgoszcz, Poland and Nicolaus Copernicus Hospital, Toruń, Poland), between 2016 and 2021, were enrolled in this study. The control group was recruited from the general population visiting Biziel University Hospital No. 2 for participation in national cancer screening programs. Patients with diagnosed serious diseases were excluded from the study. In addition, all of them had a basic cytometric test to rule out the presence of atypical cells in the peripheral blood and to assess the proportion of individual nuclear cell subfractions. The study was approved by the local Bioethics Committee at Collegium Medicum in Bydgoszcz, Nicolaus Copernicus University (KB 404/2016), and consents were obtained from the patients. All the clinical investigations were conducted according to the principles of the Declaration of Helsinki.

### 2.1. The Determination of the Epigenetic Modifications in Urine

Two-dimensional ultra-performance liquid chromatography with tandem mass spectrometry (2D UPLC–MS/MS) was used for the analysis of 5-methyl-2′-deoxycytidine (5-mdC), 5-(hydroxymethyl)-2′-deoxycytidine (5-hmdC), 5-hmCyt, 5-fCyt, 5-caCyt, and 5-(hydroxymethyl)-2′-deoxyuridine (5-hmdU). Urine samples were spiked with a mixture of stable isotope-labeled internal standards at a 4:1 volumetric ratio, and filtered before analysis. The 2D-UPLC−MS/MS system consisted of a gradient pump and autosampler for first dimension chromatography, and a gradient pump and tandem quadrupole mass spectrometer for second dimension chromatography. Both systems were coupled with a column manager equipped with two programmable column heaters and two 2-position 6-port switching valves. The following columns were used: CORTECS UPLC T3 Column (1.6 µm, 3 mm × 150 mm) with a CORTECS T3 VanGuard precolumn (1.6 µm, 2.1 mm × 5 mm) for the first dimension, a Waters ACQUITY UPLC CSH C18 (1.7 µm, 2.1 mm × 100 mm) for the second dimension, and a Waters XSelect CSH C18 column (3.5 µm, 3 mm × 20 mm) as the trap/transfer column. The chromatographic system was operated in heart-cutting mode, which means that selected portions of effluent from the first dimension were loaded onto the trap/transfer column by a 6-port switching valve, which served as an “injector” for the second dimension of the chromatography system. Mass spectrometric detection was conducted with a Waters Xevo TQ-S tandem quadrupole mass spectrometer equipped with a UniSpray ionization source as previously described by Rozalski et al. [21].

The level of 5-hmUra was determined by high-performance liquid chromatography for pre-purification followed by gas chromatography with isotope dilution mass spectrometric detection (LC/GC–MS), due to the low ionization efficiency of this molecule in UniSpray ion source [17]. In brief, urine samples enriched in labeled standard were injected onto Luna C 18(2) column (250 × 10 mm) equipped with guard column, both from Phenomenex (Torrance, CA, USA) on the HPLC system consisting of two 515 pumps, 2767 sample manager, and 2996 photodiode array detector (Waters Corp., Milford, MA, USA). The effluent was monitored with UV detector at 220–360 nm. The collected fractions containing 5-hmUra were dried by evaporation under reduced pressure in a Speed–Vac system (Thermo-Savant, Holbrook, NY, USA) and prepared for gas chromatography–mass spectrometry analysis. GC/MS analysis was performed according to the method described by Skalska et al. [22]. Excretion of urinary epigenetic modifications was estimated relative to creatinine.

A detailed description of chromatographic conditions, acquisition parameters, and data analysis is provided in the Appendix A.

### 2.2. Isolation of DNA and the Determination of the Epigenetic Modifications in DNA Isolates

Leukocytes were isolated from heparinized blood samples with Histopaque 1119 (Sigma Aldrich, St. Louise, MO, USA) solution, according to the manufacturer’s instructions, and stored at −80 °C until analysis. The analyses were performed using a method described earlier by Gackowski et al. and Starczak et al. [23,24]. Briefly, a pellet of frozen cells was dispersed in ice-cold buffer B (Tris-HCl (10 mmol/L), Na_2_EDTA (5 mmol/L), and deferoxamine mesylate (0.15 mmol/L), pH 8.0). SDS solution was added (to a final concentration of 0.5%), and the mixture was gently mixed using a polypropylene Pasteur pipette. The samples were incubated at 37 °C for 30 min. Proteinase K was added to a final concentration of 4 mg/mL and incubated at 37 °C for 1.5 h. The mixture was cooled to 4 °C, transferred to a centrifuge tube with phenol:chloroform:isoamyl alcohol (25:24:1), and vortexed vigorously. After extraction, the aqueous phase was treated with a chloroform:isoamyl alcohol mixture (24:1). The supernatant was treated with two volumes of cold absolute ethanol to precipitate high molecular weight nucleic acids. The precipitate was removed with a plastic spatula, washed with 70% (*v*/*v*) ethanol and dissolved in Milli-Q grade deionized water. The samples were mixed with 200 mM ammonium acetate containing 0.2 mM ZnCl_2_, pH 4.6 (1:1 *v*/*v*). Nuclease P1 (100 U, New England Biolabs, Ipswich, MA, USA) and tetrahydrouridine (10 μg/sample) were added to the mixture and incubated at 37 °C for 3 h. Subsequently, 10% (*v*/*v*) NH_4_OH and 6 U of shrimp alkaline phosphatase (rSAP, New England Biolabs, Ipswich, MA, USA) were added and samples were incubated for 1.5 h at 37 °C. Finally, all the hydrolysates were ultra-filtered prior to injection. The DNA hydrolysates were spiked with a mixture of internal standards at a volumetric ratio of 4:1 to a final concentration of 50 fmol/µL: [D_3_]-5-hmdC, [^13^C_10_, ^15^N_2_]-[^13^C_10_, ^15^N_2_]-5-formyl-2′-deoxycytidine (5-fdC), [^13^C_10_, ^15^N_2_]-5-carboxy-2′-deoxycytidnie (5-cadC), and [^13^C_10_, ^15^N_2_]-5-hmdU. Chromatographic separation was performed with a Waters ACQUITY 2D-UPLC system with a photodiode array detector for the first dimension of the chromatography, used for quantification of the unmodified deoxynucleosides (dN, calculated as a doubled sum of 2′-deoxythymidine and 2′-deoxyguanosine dN = 2 × (dT + dG)), and 5-mdC; and a Xevo TQ-XS tandem quadrupole mass spectrometer (used to analyze 5-hmdC, 5-fdC, 5-cadC and 5-hmdU). A detailed description of chromatographic conditions, acquisition parameters, and data analysis is provided in the Appendix A.

### 2.3. Statistical Analysis

The results are presented as mean and median values, standard deviation, interquartile ranges, and non-outlier ranges. Statistical analyses were carried out with Statistica 13.1 PL software (TIBCO Software Inc. (Palo Alto, CA, USA) (2017), Statistica (data analysis software system) version 13 (http://statistica.io, accessed on 1 March 2022), and IBM Statistics 27 PL included in PS IMAGO PRO 7.0. Parametric distribution of the variables was assessed with the Kolmogorov–Smirnov test with Lilliefors correction and based on the visual inspection of plotted histograms. Variables with parametric distribution were analyzed as raw data, while variables with non-parametric distribution were subjected to Box–Cox transformation before further analyses based on parametric tests. The one-way Student’s *t*-test (two-tailed) or LSD post hoc test was used for between-groups comparisons. All the significant differences were confirmed with the non-parametric U Mann–Whitney test.

Sensitivities, specificities, accuracy, positive and negative predictive values, and areas under receiver operating characteristic curves (ROC AUC) were calculated to compare the diagnostic value of the raw data of the analyzed parameters versus gold standard diagnostics (based on the bone marrow cytological examination). Cut-off values were determined using Youden’s index. The results were considered statistically significant at *p* values less than 0.05.

Predictors significant in the ROC analysis were included in the multifactorial classification trees model, built using exhausting Chi-squared Automatic Interaction Detector (CHAID). Input parameters for the model building were as follows: 5-times cross-validation, maximum tree depth equal to 5, minimum 4 observations in the node, Chi-square statistics based on the likelihood ratio, 10 intervals per continuous variable, and Bonferroni’s corrected *p*-value below 0.05 considered as significant.

## 3. Results

A total of 65 Caucasians (male 49%, female 51%; median age 61; range years 18–88) with primary AML and 44 Caucasians with MDS (male 52%, female 48%; median age 73 years; range years 20–87), who were newly diagnosed, treatment-naïve patients were enrolled in the study. The control group consisted of 50 Caucasian adults (median age 53 years, 42% male and 58% female, range years 33–71). Detailed patient characteristics can be found in Table 1.

Basal levels of 5-mCyt and intermediates of active demethylation pathway were analyzed in the genomic DNA isolated from the peripheral blood of treatment-naïve patients. Results are presented in Table 2 and Figure 1. The level of 5-mdC in DNA was lowest in MDS patients, reaching the statistical significance against the control group only (Figure 1a; 8.354 ± 0.491 vs. 8.594 ± 0.322 per 10^3^ dN). MDS and AML patients revealed a lower content of 5-hmdC than the control group (Figure 1b; 0.061 ± 0.037 and 0.038 ± 0.023 vs. 0.083 ± 0.028 per 10^3^ dN, respectively), and this modification was also lower in the MDS than the AML group. A gradual increase was observed in the line of the control group, MDS and AML, in the content of 5-fdC (Figure 1c; 0.136 ± 0.057, 0.208 ± 0.131, and 0.327 ± 0.253 per 10^6^ dN, respectively) and 5-cadC (Figure 1d; 9.234 ± 7.16, 17.446 ± 25.789, and 27.371 ± 25.802 per 10^9^ dN, respectively). No difference was observed for 5-hmdU in DNA (Figure 1e).

The excretion rates of intermediates of the active demethylation pathway were analyzed in spot urine and corrected to the creatinine concentration to compensate for intra-individual differences concerning the urine concentration. Results are presented in Figure 2 and Table 2. The AML patients were characterized by the highest excretion rate of all modified deoxynucleosides and nucleobases except 5-mdC, which was only higher in MDS patients than in the control group (Figure 2a; 1.663 ± 3.339 vs. 0.734 ± 0.623 nmol/mmol creatinine). 5-HmdC excretion rate reached similar values in the AML and MDS subjects, however, the values were higher than in the control one (Figure 2b; 7.338 ± 13.642 and 4.377 ± 4.228 vs. 2.136 ± 0.869 nmol/mmol creatinine, respectively). The 5-hmCyt excretion presented the same pattern as 5-caCyt and 5-hmdU gradually rising from the control group to MDS and AML (Figure 2c,e,f; 5-hmCyt: 2.55 ± 0.865, 6.307 ± 6.906, 14.594 ± 23.616; 5-caCyt: 3.397 ± 3.275; 4.918 ± 3.226, 14.403 ± 18.667; 5-hmdU: 9.223 ± 5.956, 13.399 ± 9.518, 33.514 ± 50.361 nmol/mmol creatinine, respectively). The 5-fCyt and 5-hmUra excretion in the AML subjects was higher than in the control group (Figure 2d,g; 3.858 ± 5.936 vs. 2.174 ± 0.789 and 19.84 ± 33.341 vs. 7.227 ± 1.746 nmol/mmol creatinine, respectively), but it did not differ from the MDS patients.

As the MDS and AML patients demonstrated specific patterns of the genomic content of intermediates of the active demethylation process and huge differences were observed in their urinary excretion rates, we wondered whether the analyzed compounds may serve as low-invasive biomarkers of the disease development or progression. We plotted the receiver operating characteristic (ROC) curves and calculated the area under the curve (AUC), which allowed the assessment of the biomarker utility in a manner which was independent of ad hoc choices of the thresholds. In the next step, for factors reaching the statistical significance in the ROC analysis, optimal cut-off values were determined based on the Youden’s index, assuring a balance between specificity and sensitivity. The best biomarkers allowing to distinguish between the AML patients and the control group were urinary excretion rates of 5-hmCyt, 5-hmdU, and 5-caCyt reaching AUC values of 0.918, 0.873, and 0.867 (fractions of 1 indicating “ideal” biomarker). 5-HmCyt threshold of 3.894 nmol/mmol creatinine yielded a sensitivity of 85% and a specificity of 97%. Using 5-hmdU and a threshold of 15.063 nmol/mmol creatinine yielded a sensitivity of 74% and a specificity of 92%, while using 5-caCyt and a threshold of 3.366 nmol/mmol creatinine yielded a sensitivity of 89% and a specificity of 78%. A detailed analysis is presented in Table 3 and depicted in Figure 3.

To check whether a combination of biomarkers may give a better separation power we built a classification tree including in the model all the factors significant in the univariate ROC analyses. A classification tree is a form of supervised machine learning, where the data are continuously split, according to a certain parameter, into subsets, which then split repeatedly into even smaller subsets, and so on and so forth. The process stops when the algorithm determines the data within the subsets are sufficiently homogenous or have met another stopping criterion (described in detail in Section 2.3). Using such an approach, we were able to select the most significant combination of factors allowing the correct separation of the subgroups on the basis of determined, specific cut-off values, and to present it in a human-friendly way. Four independent variables were identified (Figure 4; urinary 5-hmCyt and 5-caCyt, and 5-hmdC and 5-fdC in DNA) allowing a correct classification of 90% of the control subjects and all the AML patients (the overall prediction accuracy 95.7%).

The most significant biomarkers allowing the separation of MDS and control groups in the univariate analysis were 5-hmdC in the DNA and urinary 5-hmCyt with AUC values of 0.905 and 0.746. 5-hmdC threshold of 0.05 per 10^3^ dN yielding a sensitivity of 82% and a specificity of 98%, while using urinary 5-hmCyt and a threshold of 2.555 nmol/mmol creatinine yielded a sensitivity of 63% and a specificity of 79% (Table 4 and Figure 5). A combined multivariate separation tree identified those biomarkers as independent (Figure 6), allowing a correct classification of 98% of the control subjects and 90.9% of the MDS patients (the overall prediction accuracy 94.7%).

The power of the separation of AML from MDS was slightly lower, 5-hmCyt threshold of 8.042 nmol/mmol creatinine yielded a sensitivity of 51% and a specificity of 92%, while 5-hmdU and a threshold of 16.933 nmol/mmol creatinine yielded a sensitivity of 63% and a specificity of 84% (Figure 7 and Table 5; AUC = 0.729 and 0.749, respectively). The multi-factor approach identified two independent biomarkers (5-hmdC and 5-cadC in the DNA, allowing a correct classification of 79.5% of the MDS and 73.8% of the AML patients (the overall prediction accuracy 76.1%) (Figure 8).

As, during the study follow-up (1–5 years), some patients transformed from MDS do AML, we wondered whether any of the parameters analyzed at the time of the initial diagnosis of MDS may predict the adverse course of the disease. Despite the low number of observations, we were able to identify significant predictors of the MDS-AML transformation (Table 6 and Figure 9). The most promising are 5-hmdU, 5-cadC, and 5-hmdC in the DNA (AUCs: 0.872, 0.823, 0.811), and urinary 5-caCyt (AUC = 0.809). Unfortunately, the number of observations was not sufficient to build a classification tree based on the CHAID model.

Next, we analyzed prognostic power with the primary AML outcome–response to first-line treatment. We only noted slightly higher levels of 5-hmCyt in DNA of complete-responders (0.066/10^3^ dN) vs. partial-responders (0.039/10^3^ dN, *p* = 0.0295) and no-responders (0.05/10^3^ dN *p* = 0.0665). Moreover, 5-hmCyt in DNA was identified as a very weak predictor of complete response in ROC analysis (AUC = 0.649, *p* = 0.0159, cut-off 0.054, sensitivity 66.7%, and specificity 60.5%).

Next, we wondered whether there is an association of the analyzed intermediates of the active demethylation process with parameters currently used for patients’ risk stratification. Interestingly, we found that, in the merged MDS/AML group, high cytogenetic risk patients are characterized with higher levels of 5-hmdC and 5-fdC in DNA, and 5-hmdC and 5-hmUra in urine, than the low-risk group (Figure 10). In the same group, significant positive correlations were also found between bone marrow blast count and 5-hmdC in DNA (r = 0.5643, *p* < 0.0001) and urinary 5-hmCyt (r = 0.4641, *p* = 0.004), 5-caCyt (r = 0.7189, *p* < 0.0001), and 5-hmdU (r = 0.5646, *p* < 0.0001) (Figure 11). Consequently, in the group of MDS patients, we noted trends of positive associations of IPSS-R risk stratification and 5-hmdC in DNA and urinary 5-hmCyt, 5-caCyt, and 5-hmdU (Figure 12).

## 4. Discussion

TET2 protein mutations that compromise their catalytic activity are observed in the MDS (30–50%) as well as in the AML (30%) [25,26] patients. As mentioned in the Introduction, TET proteins convert 5-mCyt to the spectrum of epigenetic DNA modifications. Products of this process can be used to assess the extent to which TET2 mutations are responsible for the impairment of the TET2 demethylating activity. In concordance with the previous studies, we observed a significant decrease in the 5-hmdC levels in the genome of the MDS and AML patients compared to healthy subjects (Figure 1b) [20]. Surprisingly, levels of two other TETs products, i.e., 5-fdC and 5-cadC, increased significantly in both patient groups (Figure 1c,d). There was a high degree of homology within the catalytic domains of all the three TET proteins [27,28]. Therefore, it is likely that in the cases of the TET2 mutation/inactivation compensatory overexpression of the TET1/3 enzymatic activity, may restore/improve hydroxymethylation, and potentially reverse the epigenetic consequences caused by the TET2 deficiency, in the case of 5-fdC and 5-cadC. Indeed, just recently it was found that MDS patients reveal the decrease in the TET2 expression (and 5-hmdC level), while TET3 was up-regulated and inversely correlated with the TET2 expression [20].

Given the above findings, a question arises: why were significantly higher levels of 5-fCyt and 5-caCyt observed in the patient groups, while 5-hmCyt levels substantially decreased? Although the involvement of TET proteins in the generation of all the analyzed modifications is not controversial, the regulation of this process still is not fully understood. In particular, it is not clear why the oxidation of 5-mCyt once finishes at the 5-hmCyt step or progresses to the 5-fCyt and 5-caCyt. One of the reasons may be differences in the affinity of TET for 5-mCyt, 5-hmCyt, and 5-fCyt (for a review, see [29]). Additionally, various proteins modified their bases and defined their fate [30].

Although all of the paralogs harbor the same catalytic activity, they exhibit different expression patterns, which, in turn, suggests their distinct biological functions and different patterns of demethylation products in various tissues [31]. Furthermore, it was shown that the majority of 5-hmCyt, 5-fCyt, and 5-caCyt exist in the cellular DNA as stable marks [32,33]. Moreover, many reader proteins can recognize in the DNA each epigenetic mark. Among them, there are various glycosylases, DNA repair proteins, chromatin regulators, and transcription factors [34]. This strongly suggests that TET proteins, actively participating in the active demethylation process, may also individually deposit 5-hmCyt, 5-fCyt, and 5-caCyt (reviewed in [35]), which in turn may, at least partially, explain the different pattern of DNA epigenetic modifications observed in the patient groups.

Many studies demonstrated the utility of DNA epigenetic modifications to discriminate cancerous cells. However, the scarcity of works concerning urinary epigenetic modifications justifies the question of why the urinary exertion rate of epigenetically modified bases or nucleosides plays a similar role. The simple answer is the origin of urinary modifications. Both the active DNA demethylation and DNA repair are involved in the removal of 5-fCyt and 5-caCyt, which are replaced with unmodified cytosine. The presence of 5-fCyt and 5-caCyt in the DNA can inhibit the DNA replication, leading to the genomic instability [36,37]. Therefore, thymine DNA glycosylase, which has a strong excision activity against 5-fCyt and 5-caCyt (TDG), is responsible for removing these modifications from the DNA [38,39,40]. The removed modifications in the form of bases or deoxynucleosides are released into the bloodstream and are eventually excreted with the urine [18]. The main source of modifications in the analyzed urine are thus the DNA repair mechanisms equipped with efficient enzymatic systems.

The processive demethylation pathway described in the earlier studies may be another source of epigenetic markers excreted in urine [41]. The presence of 5-fCyt, 5-caCyt, and 5-hmUra in the DNA leads to the initiation of the processive DNA demethylation and consequently induces the demethylation of many 5-mCyts (and possibly 5-hmCyts) at the same locus via the long-patch BER pathway, the nucleotide excision repair, or the DNA mismatch repair. Recent experiments demonstrated that the presence of 5-hmUra can initiate the removal of distant epigenetic modifications (5-mCyt and 5-hmCyt) via mismatch repair and long-patch BER pathways, which may justify the urinary excretion of 5-hmCyt and 5-mCyt deoxynucleosides [42].

It is estimated that approximately 30% of MDS patients progress to AML [43]. Disturbed epigenetic processes are often involved in the evolution and progression of MDS to AML. This notion is supported by the studies evincing that, in MDS and AML, among the most commonly mutated genes are those affected by the factors responsible for the epigenetic regulation [44]. Moreover, epigenetic aberrations coexist with other genetic abnormalities in MDS and AML and together cause the development of the disease [45]. It is therefore suggested that the accumulation of epigenetic changes is a significant factor in the transformation of MDS to AML [46]. Moreover, mutations of TET2 and IDH1/2 (both the enzymes are involved in shaping DNA epigenetic modifications) belong to the driving mutations that are acquired with the evolution of MDS to AML [47]. While TET2 mutations cause only changes in TET2-specific products, IDH mutations, producing oncometabolite 2-hydroxyglutarate, result in the inhibition of the whole TET enzymes family, as well as other 2-oxyglutarate-dependent dioxygenases [48]. It may be one of the reasons for a different pattern of TET products observed in AML and MDS.

Patients whose MDS evolves often will require more aggressive therapy. Identifying them early can help determine when to start aggressive therapy. Therefore, it is extremely important to find biomarkers for people with MDS who have a significantly increased incidence of evolution into AML. For this purpose we used the calculated level of the epigenetic modification, both analyzed in the cellular DNA and excreted into the urine, to construct ROC curves and to build up multi-factor models of classification trees. Receiver operating characteristic curves demonstrated their diagnostic availability to discriminate AML and MDS from controls (Figure 3 and Figure 5). The curves were also helpful for the separation between MDS and AML (Figure 7). Moreover, and importantly, the multi-factor model identified the most significant independent variables and showed the high diagnostic value of the analyzed parameters in distinguishing AML (Figure 4) and MDS (Figure 6) patients from the control group, as well as between both diseases (Figure 8).

The most diagnostically useful parameter for discriminating AML patients from the control group was the urinary excretion of 5-hmCyt and 5-hmdU, and for MDS patients, 5-hmdC in DNA and urinary 5-hmCyt. Multi-factor models of classification trees allowed the correct classification of patients with AML and MDS in 95.7% and 94.7% of cases, respectively. The highest prognostic value of the analyzed parameters for predicting the transformation of MDS into AML have 5-cadC and 5-hmdU in DNA.

The presented research proves that the intermediates of the active DNA demethylation pathway determined in the completely non-invasive (urine) or minimally invasive (blood) material can be useful in supporting the diagnostic process of patients with MDS and AML. The possibility of an early identification of a group of MDS patients with an increased risk of transformation into AML is of particular importance. The low invasiveness of the proposed determinations makes them particularly useful in the routine monitoring of MDS patients, bringing another layer of information in addition to currently used genetic testing.

## Figures and Tables

**Figure 1 cells-11-00888-f001:**
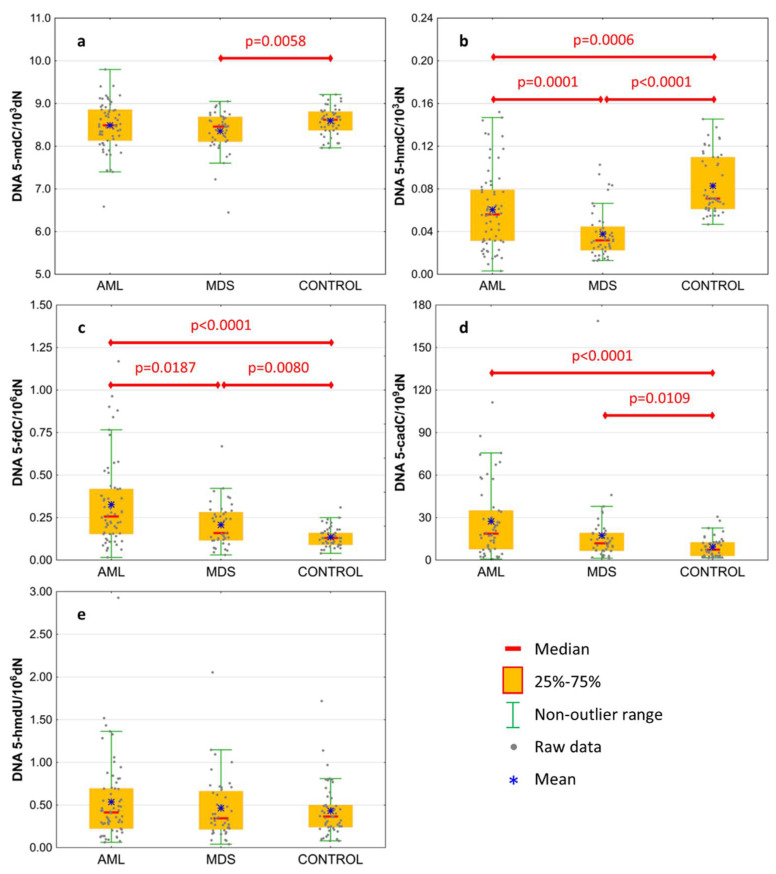
Levels of active demethylation products in leukocyte DNA from patients with acute myeloid leukemia (AML), myelodysplastic syndromes (MDS), and healthy controls (CONTROL). (**a**) 5-methyl-2′-deoxycytidine (5-mdC); (**b**) 5-(hydroxymethyl)-2′-deoxycytidine (5-hmdC); (**c**) 5-formyl-2′-deoxycytidine (5-fdC); (**d**) 5-carboxyl-2′-deoxycytidine (5-cadC); (**e**) 5-(hydroxymethyl)-2′-deoxyuridine (5-hmdU). Detailed analysis of results to be found in Table 2.

**Figure 2 cells-11-00888-f002:**
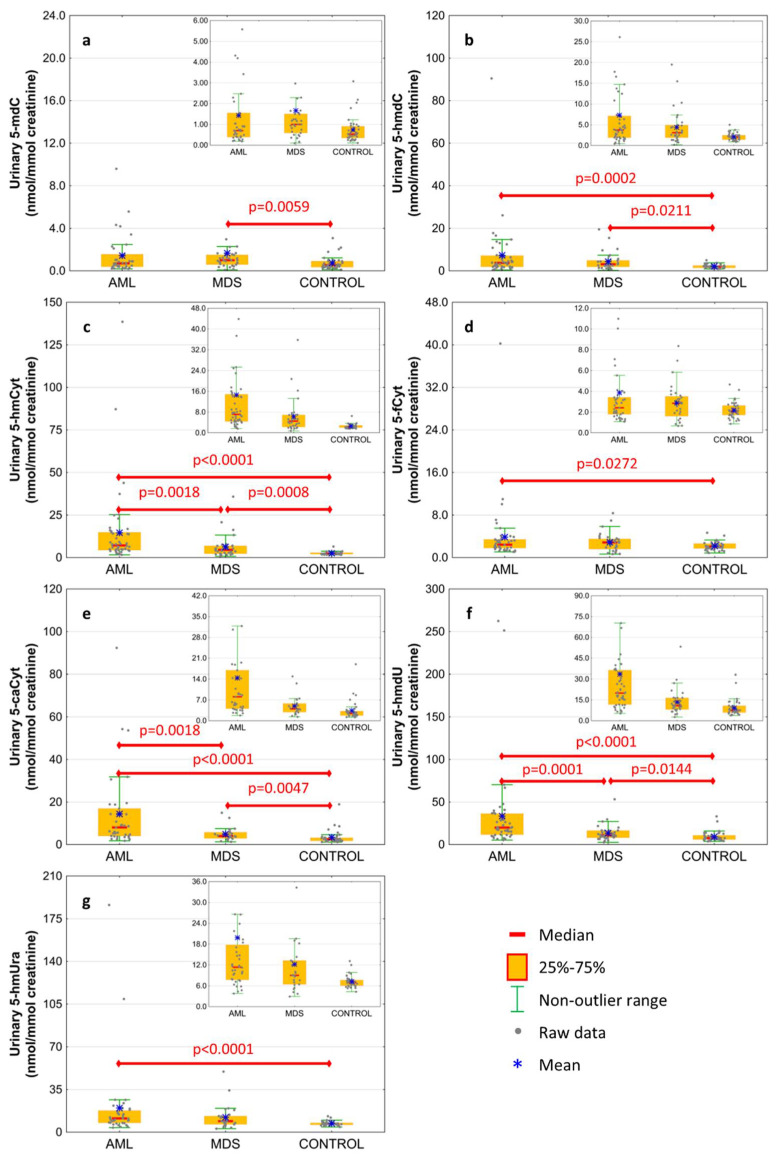
Levels of active demethylation products in urine from patients with acute myeloid leukemia (AML), myelodysplastic syndromes (MDS) and healthy controls (CONTROL). (**a**) 5-methyl-2′-deoxycytidine (5-mdC); (**b**) 5-(hydroxymethyl)-2′-deoxycytidine (5-hmdC); (**c**) 5-hydroxymethylcytosine (5-hmCyt); (**d**) 5-formylcytosine (5-fCyt); (**e**) 5-carboxycytosine (5-caCyt); (**f**) 5-(hydroxymethyl)-2′-deoxyuridine (5-hmdU); (**g**) 5-hydroxymethyluracil (5-hmUra). Detailed analysis of results to be found in Table 2.

**Figure 3 cells-11-00888-f003:**
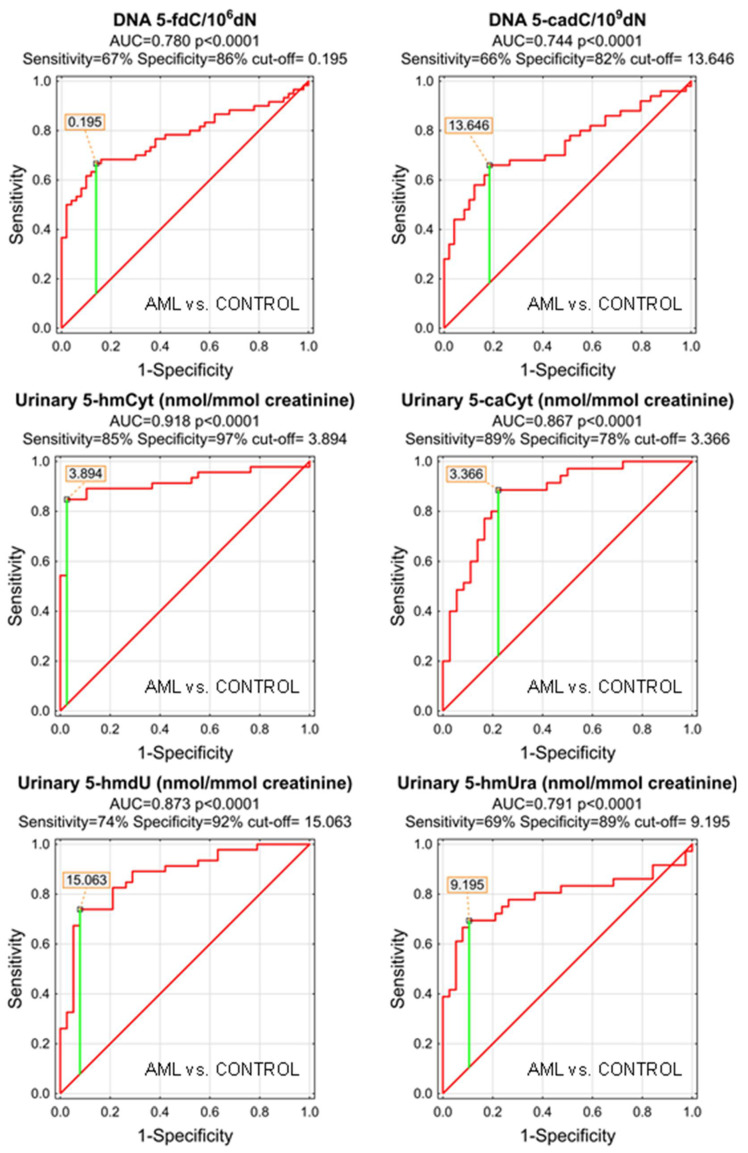
Most significant Receiver Operating Characteristic (ROC) curves (AML vs. CONTROL) for epigenetic modification in the leukocyte DNA and urine. AUC-area under the curve. Detailed analysis of results is in Table 3.

**Figure 4 cells-11-00888-f004:**
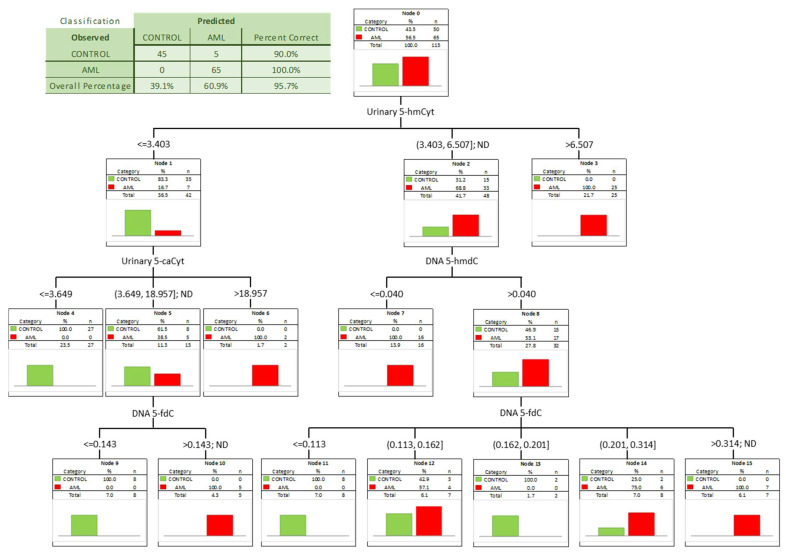
Classification tree for distinguishing acute myeloid leukemia patients (AML) from healthy controls (CONTROL).

**Figure 5 cells-11-00888-f005:**
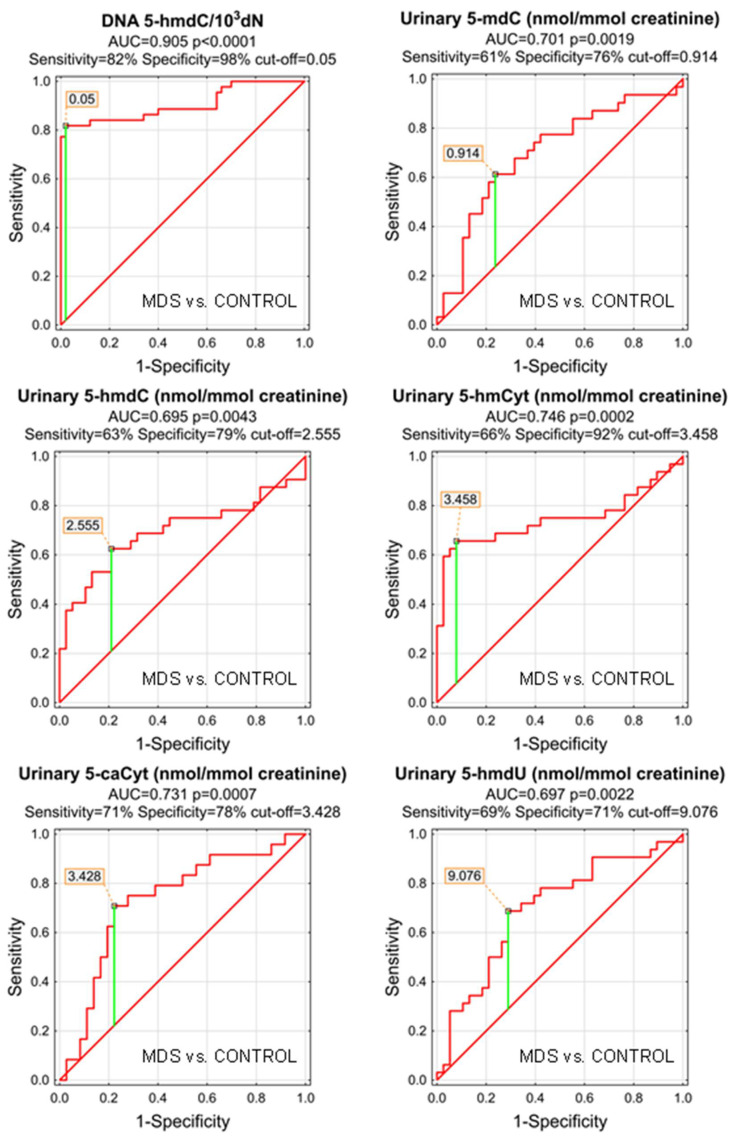
Most significant Receiver Operating Characteristic (ROC) curves (MDS vs. CONTROL) for the epigenetic modification in the leukocyte DNA and urine. AUC-area under the curve. Detailed analysis of results to be found in Table 4.

**Figure 6 cells-11-00888-f006:**
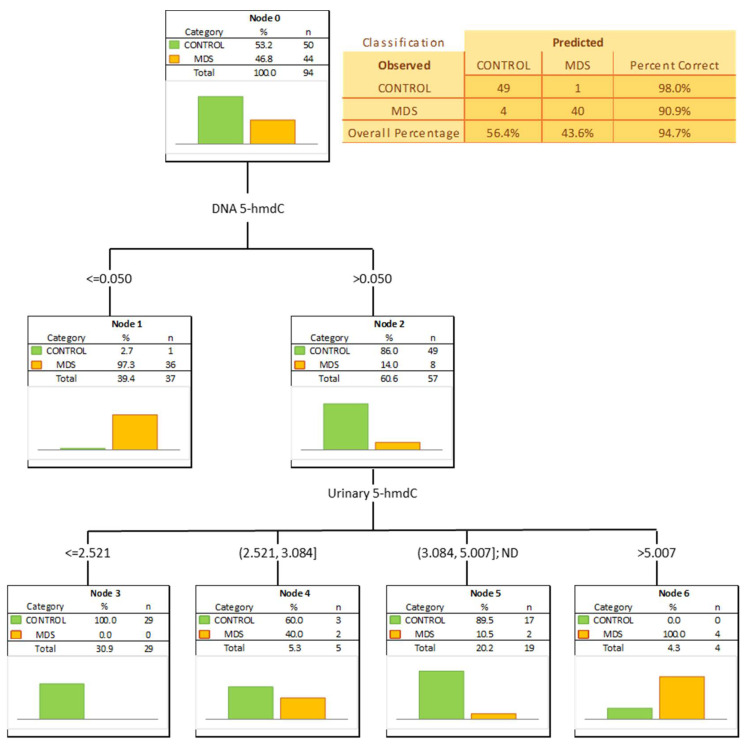
Classification tree distinguishing myelodysplastic syndromes patients (MDS) from healthy controls (CONTROL).

**Figure 7 cells-11-00888-f007:**
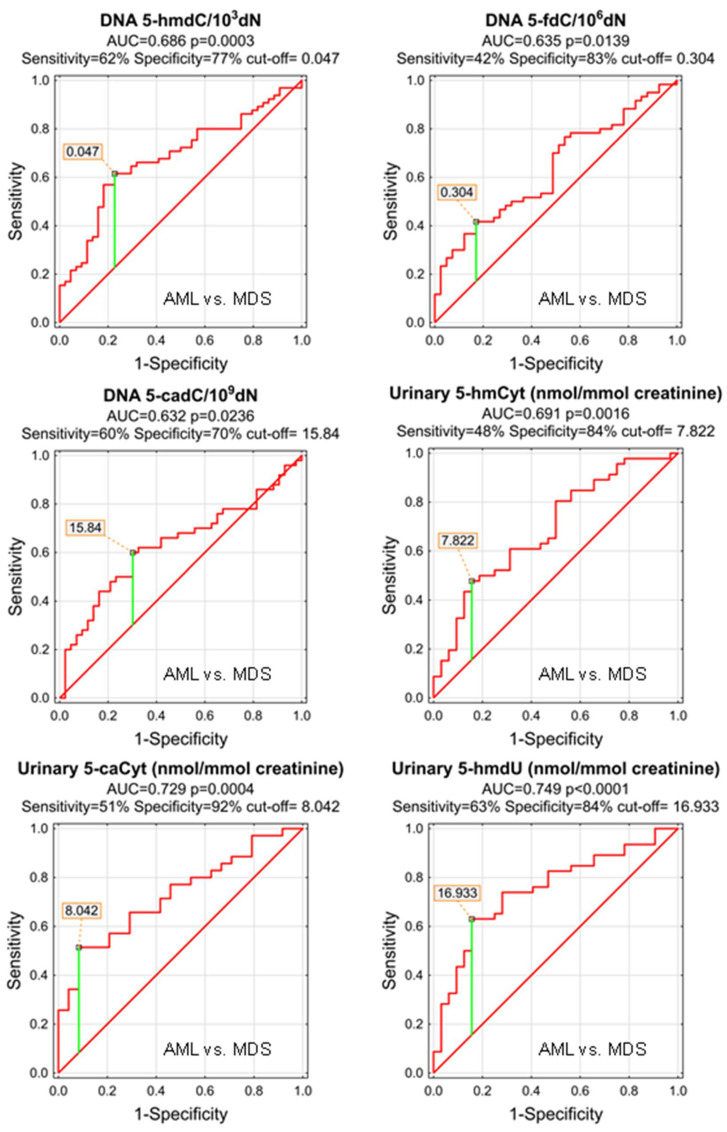
Most significant Receiver Operating Characteristic (ROC) curves (AML vs. MDS) for the epigenetic modification in the leukocyte DNA and urine. AUC-area under the curve. Detailed analysis of results to be found in Table 5.

**Figure 8 cells-11-00888-f008:**
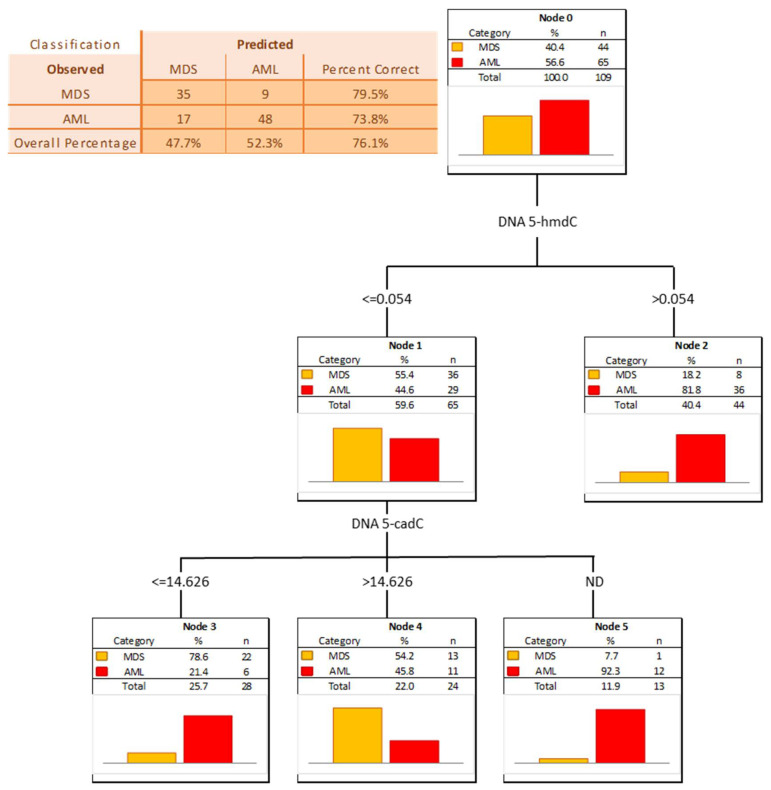
Classification tree for distinguishing acute myeloid leukemia patients (AML) from myelodysplastic syndromes patients (MDS).

**Figure 9 cells-11-00888-f009:**
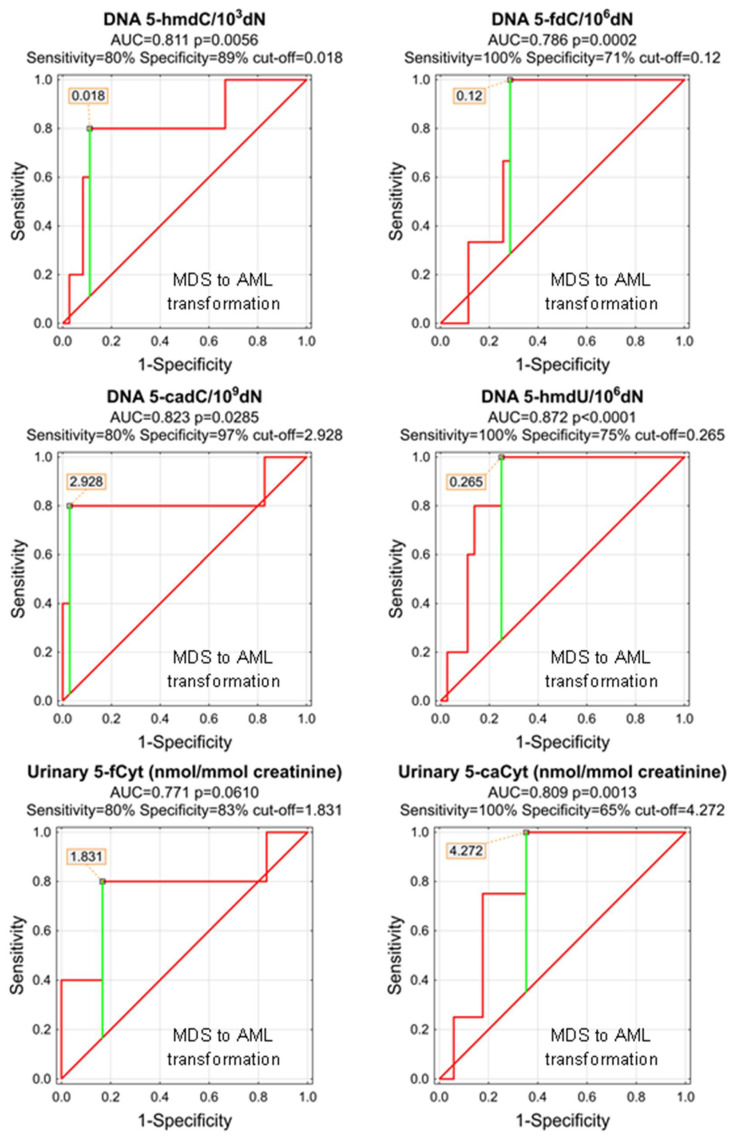
Most significant Receiver Operating Characteristic (ROC) curves (MDS to AML transformation) for the epigenetic modification in the leukocyte DNA and urine. AUC-area under the curve. Detailed analysis of results to be found in Table 6.

**Figure 10 cells-11-00888-f010:**
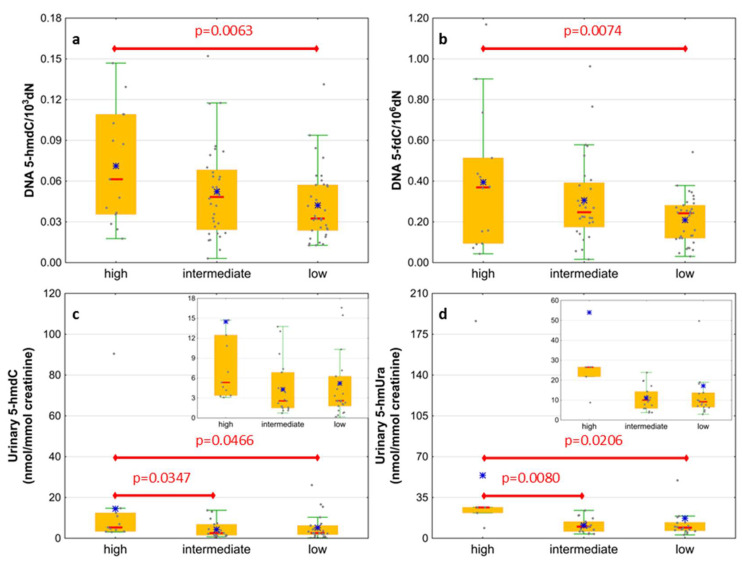
Comparison of the levels of selected epigenetic modifications in leukocyte DNA and urine between the cytogenetic risk groups. (**a**) 5-(hydroxymethyl)-2’-deoxycytidine (5-hmdC) in DNA; (**b**) 5-formyl-2’-deoxycytidine (5-fdC) in DNA; (**c**) 5-(hydroxymethyl)-2’-deoxyuridine (5-hmdU) in urine; (**d**) 5-hydroxymethyluracil (5-hmUra) in urine.

**Figure 11 cells-11-00888-f011:**
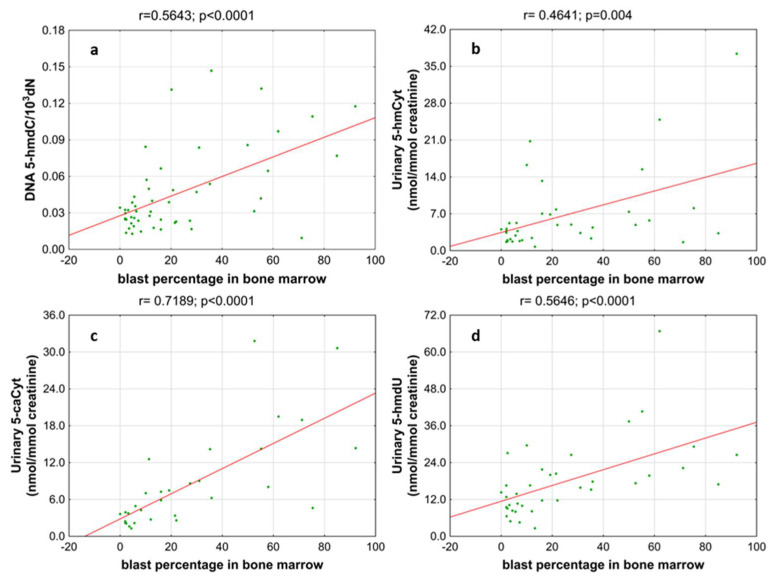
Most significant correlations between the blast percentage in bone marrow and epigenetic markers in the leukocyte DNA and urine. (**a**) 5-(hydroxymethyl)-2’-deoxycytidine (5-hmdC) in DNA; (**b**) 5-hydroxymethylcytosine (5-hmCyt) in urine; (**c**) 5-carboxycytosine (5-caCyt) in urine; (**d**) 5-(hydroxymethyl)-2’-deoxyuridine (5-hmdU) in urine.

**Figure 12 cells-11-00888-f012:**
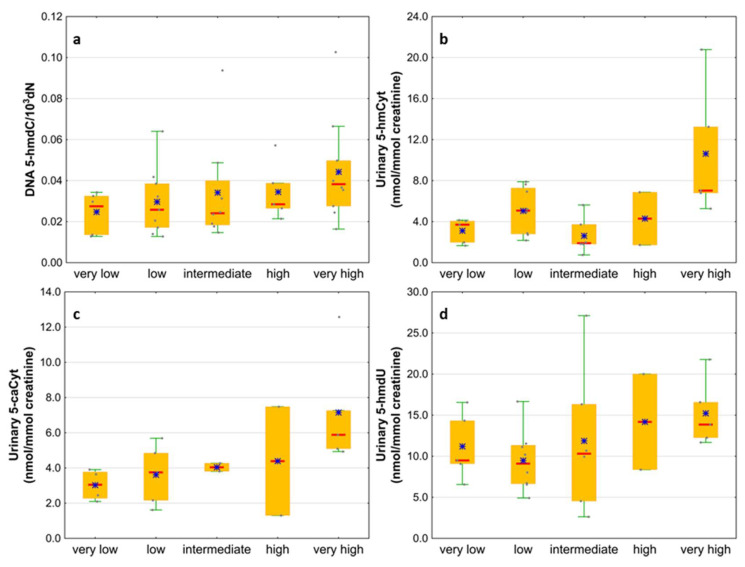
Comparison of the levels of selected epigenetic modifications in leukocyte DNA and urine between the IPSS-R risk groups. (**a**) 5-(hydroxymethyl)-2’-deoxycytidine (5-hmdC) in DNA; (**b**) 5-hydroxymethylcytosine (5-hmCyt) in urine; (**c**) 5-carboxycytosine (5-caCyt) in urine; (**d**) 5-(hydroxymethyl)-2’-deoxyuridine (5-hmdU) in urine.

**Table 1 cells-11-00888-t001:** Detailed characteristic of the patients’ groups.

		Number of Patients, n (%)
MDS patients		44
Age	<60 years	8 (19)
	≥60 years	34 (81)
Sex	Male	24 (55)
	Female	20 (45)
BM blast count	<5%	12 (39)
	5–10%	8 (26)
	11–20%	11 (35)
WHO subtype	RA	1 (2)
	RARS	1 (2)
	RCMD	10 (22)
	‘5q−syndrome’	1 (2)
	RAEB-1	7 (17)
	RAEB-2	14 (35)
	MDS-U	9 (20)
IPSS-R risk group	Very low	6 (16)
	Low	9 (24)
	Intermediate	8 (21)
	High	5 (13)
	Very high	10 (26)
Cytogenetic risk group	Low	25 (71)
	Intermediate	5 (14)
	High	5 (14)
AML patients		65
Age	<60 years	24 (40)
	≥60 years	40 (63)
Sex	Male	33 (51)
	Female	32 (49)
BM blast count	20–29%	6 (32)
	>30%	13 (68)
Cytogenetic risk group	Low	8 (18)
	Intermediate	27 (61)
	High	9 (20)

Abbreviations: BM, bone marrow; IPSS-R, Revised International Prognostic Scoring System; MDS-U, MDS unclassifiable; RA, refractory anemia; RAEB, RA with excess of blasts; RARS, RA with ringed sideroblasts; RCMD, refractory cytopenia with multilineage dysplasia; and WHO, World Health Organization.

**Table 2 cells-11-00888-t002:** Comparison of active demethylation pathway intermediates in the study groups.

		AML	MDS	CONTROL	AML vs. CONTROL	MDS vs. CONTROL	AML vs. MDS
		Mean ± SD			*t*-Student Test		
		Median (Interquartile Range)			U-Mann–Whitney Test		
DNA	5-methyl-2′-deoxycytidine	8.49 ± 0.56	8.354 ± 0.491	8.594 ± 0.322	0.2148	0.0058	0.2018
		8.49 (8.132;8.859)	8.459 (8.106;8.687)	8.622 (8.372;8.805)	0.3446	0.0190	0.2096
DNA	5-(hydroxymethyl)-2′-deoxycytidine	0.061 ± 0.037	0.038 ± 0.023	0.083 ± 0.028	0.0006	<0.0001	0.0001
		0.056 (0.031;0.079)	0.032 (0.023;0.045)	0.071 (0.061;0.11)	0.0002	<0.0001	0.0010
DNA	5-formyl-2′-deoxycytidine	0.327 ± 0.253	0.208 ± 0.131	0.136 ± 0.057	<0.0001	0.0080	0.0187
		0.257 (0.154;0.419)	0.159 (0.116;0.282)	0.131 (0.094;0.164)	<0.0001	0.0130	0.0215
DNA	5-carboxy-2′-deoxycytidine	27.371 ± 25.802	17.446 ± 25.789	9.234 ± 7.16	<0.0001	0.0109	0.0618
		18.678 (7.749;34.981)	11.915 (6.587;19.176)	7.353 (2.983;12.601)	<0.0001	0.0197	0.0295
DNA	5-(hydroxymethyl)-2′-deoxyuridine	0.536 ± 0.478	0.463 ± 0.372	0.431 ± 0.306	0.3592	0.8438	0.7834
		0.414 (0.222;0.695)	0.344 (0.214;0.663)	0.363 (0.235;0.498)	0.3880	0.9728	0.4866
urine	5-methyl-2′-deoxycytidine	1.435 ± 1.936	1.663 ± 3.339	0.734 ± 0.623	0.0955	0.0059	0.3664
		0.701 (0.389;1.554)	1 (0.587;1.506)	0.534 (0.346;0.908)	0.1983	0.0043	0.2174
urine	5-(hydroxymethyl)-2′-deoxycytidine	7.338 ± 13.642	4.377 ± 4.228	2.136 ± 0.869	0.0002	0.0211	0.3459
		3.746 (1.902;7.138)	3.086 (1.917;4.913)	2.032 (1.439;2.488)	0.0005	0.0053	0.4311
urine	5-hydroxymethylcytosine	14.594 ± 23.616	6.307 ± 6.906	2.55 ± 0.865	<0.0001	0.0008	0.0018
		7.142 (4.394;14.866)	4.543 (2.288;6.901)	2.344 (2.047;2.866)	<0.0001	0.0004	0.0044
urine	5-formylcytosine	3.858 ± 5.936	2.88 ± 1.741	2.174 ± 0.789	0.0272	0.3259	0.4543
		2.419 (1.793;3.42)	2.853 (1.607;3.507)	2.062 (1.713;2.626)	0.0669	0.0554	0.8809
urine	5-carboxycytosine	14.403 ± 18.667	4.918 ± 3.226	3.397 ± 3.275	<0.0001	0.0047	0.0018
		8.042 (4.055;16.928)	3.977 (2.91;5.785)	2.453 (1.746;3.195)	<0.0001	0.0026	0.0031
urine	5-(hydroxymethyl)-2′-deoxyuridine	33.514 ± 50.361	13.399 ± 9.518	9.223 ± 5.956	<0.0001	0.0144	0.0001
		19.942 (11.7;36.332)	10.904 (8.093;16.438)	7.518 (5.931;10.78)	<0.0001	0.0049	0.0002
urine	5-hydroxymethyluracil	19.84 ± 33.341	12.197 ± 10.309	7.227 ± 1.746	<0.0001	0.0708	0.1528
		11.341 (7.718;17.778)	9.018 (6.428;13.269)	6.921 (6.052;7.658)	<0.0001	0.0160	0.1570

**Table 3 cells-11-00888-t003:** ROC analysis for AML vs. CONTROL group.

AML vs. CONTROL											
		AUC	SE	*p*	S/D	Cut-Off	Sensitivity	Specificity	Accuracy	PPV	NPV
DNA	5-methyl-2′-deoxycytidine	0.552	0.054	0.3394	D						
DNA	5-(hydroxymethyl)-2′-deoxycytidine	0.702	0.049	<0.0001	D	0.049	43%	98%	67%	97%	57%
DNA	5-formyl-2′-deoxycytidine	0.780	0.045	<0.0001	S	0.195	67%	86%	75%	85%	68%
DNA	5-carboxy-2′-deoxycytidine	0.744	0.051	<0.0001	S	13.64	66%	82%	74%	79%	70%
DNA	5-(hydroxymethyl)-2′-deoxyuridine	0.547	0.054	0.3890	S						
urine	5-methyl-2′-deoxycytidine	0.588	0.067	0.1898	S						
urine	5-(hydroxymethyl)-2′-deoxycytidine	0.723	0.057	0.0001	S	3.627	54%	95%	73%	93%	63%
urine	5-hydroxymethylcytosine	0.918	0.033	<0.0001	S	3.894	85%	97%	90%	98%	84%
urine	5-formylcytosine	0.618	0.062	0.0581	S						
urine	5-carboxycytosine	0.867	0.043	<0.0001	S	3.366	89%	78%	83%	79%	88%
urine	5-(hydroxymethyl)-2′-deoxyuridine	0.873	0.039	<0.0001	S	15.063	74%	92%	82%	92%	74%
urine	5-hydroxymethyluracil	0.791	0.058	<0.0001	S	9.195	69%	89%	80%	86%	76%

AUC—area under curve; SE—standard error; S—stimulant; D—destimulant; PPV—positive predictive value; and NPV—negative predictive value.

**Table 4 cells-11-00888-t004:** ROC analysis for MDS vs. CONTROL group.

MDS vs. CONTROL											
		AUC	SE	*p*	S/D	Cut-Off	Sensitivity	Specificity	Accuracy	PPV	NPV
DNA	5-methyl-2′-deoxycytidine	0.642	0.057	0.0133	D	8.601	67%	56%	61%	57%	67%
DNA	5-(hydroxymethyl)-2′-deoxycytidine	0.905	0.034	<0.0001	D	0.05	82%	98%	90%	97%	86%
DNA	5-formyl-2′-deoxycytidine	0.654	0.062	0.0125	S	0.228	49%	92%	73%	83%	69%
DNA	5-carboxy-2′-deoxycytidine	0.642	0.058	0.0145	S	6.497	81%	45%	62%	56%	73%
DNA	5-(hydroxymethyl)-2′-deoxyuridine	0.503	0.061	0.9551	S						
urine	5-methyl-2′-deoxycytidine	0.701	0.065	0.0019	S	0.914	61%	76%	70%	68%	71%
urine	5-(hydroxymethyl)-2′-deoxycytidine	0.695	0.068	0.0043	S	2.555	63%	79%	71%	71%	71%
urine	5-hydroxymethylcytosine	0.746	0.066	0.0002	S	3.458	66%	92%	80%	88%	76%
urine	5-formylcytosine	0.634	0.072	0.0635	S						
urine	5-carboxycytosine	0.731	0.068	0.0007	S	3.428	71%	78%	75%	68%	80%
urine	5-(hydroxymethyl)-2′-deoxyuridine	0.697	0.064	0.0022	S	9.076	69%	71%	70%	67%	73%
urine	5-hydroxymethyluracil	0.681	0.080	0.0232	S	8.666	60%	84%	75%	71%	76%

AUC—area under curve; SE—standard error; S—stimulant; D—destimulant; PPV—positive predictive value; and NPV—negative predictive value.

**Table 5 cells-11-00888-t005:** ROC analysis for AML vs. MDS group.

AML vs. MDS											
		AUC	SE	*p*	S/D	Cut-Off	Sensitivity	Specificity	Accuracy	PPV	NPV
DNA	5-methyl-2′-deoxycytidine	0.572	0.055	0.1927	S						
DNA	5-(hydroxymethyl)-2′-deoxycytidine	0.686	0.051	0.0003	S	0.047	62%	77%	68%	80%	58%
DNA	5-formyl-2′-deoxycytidine	0.635	0.055	0.0139	S	0.304	42%	83%	58%	78%	49%
DNA	5-carboxy-2′-deoxycytidine	0.632	0.058	0.0236	S	15.84	60%	70%	65%	70%	60%
DNA	5-(hydroxymethyl)-2′-deoxyuridine	0.540	0.057	0.4836	S						
urine	5-methyl-2′-deoxycytidine	0.589	0.072	0.2169	D						
urine	5-(hydroxymethyl)-2′-deoxycytidine	0.553	0.066	0.4203	S						
urine	5-hydroxymethylcytosine	0.691	0.061	0.0016	S	7.822	48%	84%	63%	81%	53%
urine	5-formylcytosine	0.510	0.068	0.8790	S						
urine	5-carboxycytosine	0.729	0.065	0.0004	S	8.042	51%	92%	68%	90%	56%
urine	5-(hydroxymethyl)-2′-deoxyuridine	0.749	0.056	<0.0001	S	16.933	63%	84%	72%	85%	61%
urine	5-hydroxymethyluracil	0.608	0.074	0.1475	S						

AUC—area under curve; SE—standard error; S—stimulant; D—destimulant; PPV—positive predictive value; and NPV—negative predictive value.

**Table 6 cells-11-00888-t006:** ROC analysis for MDS to AML transformation.

MDS to AML Transformation											
		AUC	SE	*p*	S/D	Cut-Off	Sensitivity	Specificity	Accuracy	PPV	NPV
DNA	5-methyl-2′-deoxycytidine	0.514	0.108	0.8975	S						
DNA	5-(hydroxymethyl)-2′-deoxycytidine	0.811	0.112	0.0056	D	0.018	80%	89%	88%	50%	97%
DNA	5-formyl-2′-deoxycytidine	0.786	0.076	0.0002	D	0.12	100%	71%	74%	23%	100%
DNA	5-carboxy-2′-deoxycytidine	0.823	0.147	0.0285	D	2.928	80%	97%	95%	80%	97%
DNA	5-(hydroxymethyl)-2′-deoxyuridine	0.872	0.057	<0.0001	D	0.265	100%	75%	78%	36%	100%
urine	5-methyl-2′-deoxycytidine	0.643	0.164	0.3822	D						
urine	5-(hydroxymethyl)-2′-deoxycytidine	0.583	0.161	0.6054	D						
urine	5-hydroxymethylcytosine	0.550	0.154	0.7449	D						
urine	5-formylcytosine	0.771	0.145	0.0610	D						
urine	5-carboxycytosine	0.809	0.096	0.0013	S	4.272	100%	65%	71%	40%	100%
urine	5-(hydroxymethyl)-2′-deoxyuridine	0.508	0.144	0.9539	D						
urine	5-hydroxymethyluracil	0.596	0.236	0.6825	D						

AUC—area under curve; SE—standard error; S—stimulant; D—destimulant; PPV—positive predictive value; and NPV—negative predictive value.

## Data Availability

The datasets obtained in the current study are available from the corresponding author on reasonable request.

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
