# Peer review of "Diagnostic and Prognostic Power of Active DNA Demethylation Pathway Intermediates in Acute Myelogenous Leukemia and Myelodysplastic Syndromes"

_cells, 2022, doi:10.3390/cells11050888_

Round 1

Reviewer 1 Report

In the present work the authors have investigated the diagnostic and prognostic power of active DNA demethylation pathway intermediates in acute myelogenous leukemia and myelodysplastic syndromes.

Introduction

The authors report that they analyzed "...the levels of 5-mCyt oxidation products in the minimally-invasive material: the cellular DNA from peripheral blood cells and the urine of patients...": Which products? what were the molecules under investigation?

the authors should state the scope of their work, clearly, at the end of this section and not describe their approach.

Materials and Methods

The authors should summarize the patient cohort in a Table.

In section 2.1. please describe briefly the method. It will be more easy for then non-specialist to follow the investigation by having the methodology at hand. In the "Introduction" section the authors state that they investigated 5-mCyt, while in section 2.1. they mention 5-hmUra. Please clarify.

In section 2.2. the authors mention that cells were stored at -80oC. Yet, during thawing cells rupture. How did the authors account for this detail? What is the extraction process followed? referring to a previous work is not sufficient to explain their method. They should add the info how they extracted the molecules of interest. Especially, what was the extraction method from leukocytes?

Results

In line 158 authors state that they have investigated for global methylation. How was that performed? in the "Materials and Methods" section they referred to "demethlation" this is confusing. Please explain.

Please explain the dN unit.

The presented results indicated the diagnostic power of demethylation between controls, AML, MDS. Yet, the authors refer to prognosis in their manuscript as well as in the title. How did they account for prognosis? it is necessary to present survival curves of patients, related to the demethylation status of their samples.

Please give some more detail on their approach of presenting the classification trees. How does this approach help towards the understanding of Control-MDS-AML diagnosis?

Please explain the transition steps from control to AML if any. Is there an observed mechanism indicating that from control to AML samples (with intermediate MDS) demethylation plays a role?

Discussion

The authors should highlight their findings and summarize the use of their findings for leukemia diagnosis and prognosis? is it possible to use these findings in the clinical praxis?

Reviewer 2 Report

Skalska-Bugala et al. report on blood and urine sample to detect DNA demethylation pathway intermediates as diagnostic and prognostic in AML and MDS. The paper may benefit from the following suggestions:

-Please comment on  whether these markers were also tested in the bone marrow. AML and MDS are morphological diagnosis and therefore having blood or urine markers is helpful but need to be investigated in the context of bone marrow findings. These markers should be  tested on the  marrow and compared to blood and urine samples.

-Move tables from supplementary files to main file 

-The results from the first 2 lines of methods section should be moved to results section

Reviewer 3 Report

The manuscript by Aleksandra Skalska-Bugala et al reports that quantification of DNA demethylation pathway intermediates (using 2D ultra-performance liquid chromatography with tandem mass spectrometry) in the urine or the blood could be used to support the distinction between myelodysplastic syndrome (MDS) or acute myeloid leukemia (AML) and controls as well as to predict the transformation of MDS into AML. A cut-off value is proposed for the most significant molecules, together with their ability to distinguish AML, MDS and controls.

This is an interesting study that is carefully reported but it remains unclear how useful these analyses could be as recognition of MDS and AML is based currently on a simple morphological analysis of bone marrow cells with a cut-off based on blast cell fraction, which would hardly be replaced by such analyses. Therefore, as presented, it is difficult to translate these results into clinically relevant information: does the measurement of these intermediate replace the morphological analysis of the bone marrow? No. Then the question is: what else could we learn from these data?

A suggestion could be to translate the heterogeneous amounts of the measured intermediates into a heatmap with each DNA demethylation intermediate as a line, each individual patient as a column, and the measured amounts as a range of colors. Then, a non-supervised analysis would indicate whether patient subgroups can be identified in disease groups, e.g. do MDS at high risk of transformation into AML demonstrate a distinct and characteristic pattern, in blood, in urine, or in the combination of both? Are there other subgroups that could be significant?   

Another important limitation of this study is that very few information is provided on patient diseases: is there a link between the prognostic score of MDS (according to R-IPSS for example) and the pattern of the measured intermediates? Is there a link between the nature of AML (de novo or secondary for example, WHO subgroup, pattern of somatic mutations, etc..) and the measured pattern? Is there a pattern of prognostic significance? And what are the controls (they appear to be patients with other diseases, thus a possible inflammatory state that could influence blood cell epigenetic pattern). Altogether, more details on studied patients are requested

Another point is that mutations in TET2 and IDH genes are not equal, mutations in TET2 are common in MDS, those in IDH are usually associated with AML or predict AML transformation

The last sentence is a bit provocative: how to conclude that this monitoring could replace genetic testing without comparing the reported results to genetic testing?

Round 2

Reviewer 1 Report

The authors have addressed my previous comments. Their work has merit for publication.

Reviewer 2 Report

Thank you for addressing the comments. 

Reviewer 3 Report

Thank you for carefully addressing the comments made in the initial review and modifying your manuscript accordingly.   I have no additional suggestion.